# Towards Better Evaluation Metrics for Text-to-Motion Generation

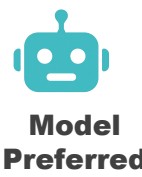 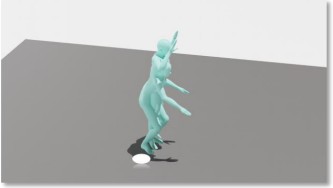 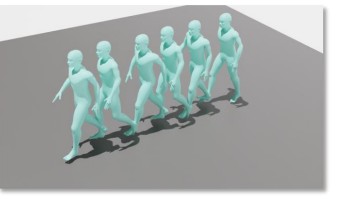 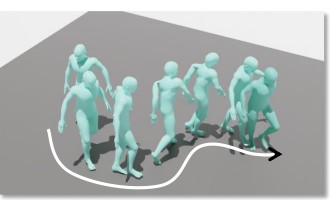

**Model Preferred**

a person **walks backward** and jumps.    a man walks forward and **picks up a box**.    a person walked and **turned right**.

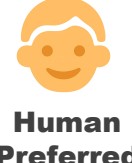 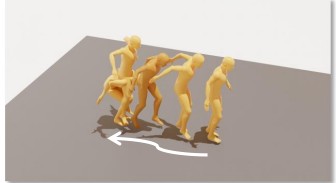 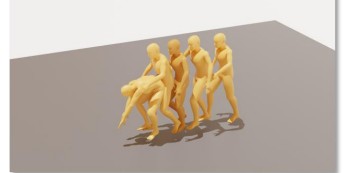 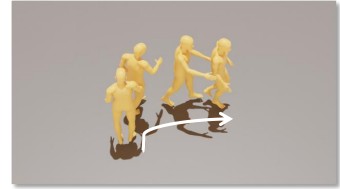

**Human Preferred**

Figure 1: Within the field of text-to-motion generation numerous contemporary models report achieving exceptionally high quantitative metrics. Some models even claim performance exceeding ground truth benchmarks. However a notable disparity often exists between these reported scores and the practical outcomes. The actual generated motions frequently exhibit poor quality and do not align well with human preferences or perceptual expectations.

## Abstract

A reliable evaluation metric is essential for guiding positive developments within a research field. In the domain of text-to-motion generation, the traditional evaluation metrics such as Fréchet Inception Distance (FID) and R-Precision suffer from inherent limitations. Specifically, FID is biased by its Gaussian assumption, while R-Precision lacks global awareness. Current work often overemphasizes improvements on these unreliable metrics to indicate model superiority. To address these challenges, we propose two novel evaluation metrics: Optimal Transport Matching Score (OTMS) and MoCLIP-based Maximum Mean Discrepancy (MMMD). OTMS formulates text-motion matching as an optimal transport process, enabling a global perspective. MMMD leverages our enhanced Mo-CLIP encoder and Gaussian-RBF-based Maximum Mean Discrepancy, providing an unbiased evaluation without restrictive distribution assumptions. Extensive experiments and analysis demonstrate that our proposed metrics align closely with human perceptual judgments and provide efficient, comprehensive, and reliable evaluations for text-driven motion generation tasks. The code can be found on the anonymous website.

## CCS Concepts

• **Computing methodologies** → **Computer vision**.

## Keywords

Text-to-Motion Generation, Optimal Transport, Evaluation Metrics

**ACM Reference Format:**
Anonymous Author(s). 2018. Towards Better Evaluation Metrics for Text-to-Motion Generation. In *Proceedings of Make sure to enter the correct conference title from your rights confirmation email (Conference acronym 'XX)*. ACM, New York, NY, USA, 11 pages. https://doi.org/XXXXXXX.XXXXXXX

## 1 Introduction

"The measure of intelligence is the ability to change." – Albert Einstein

Text-to-motion synthesis has witnessed rapid progress, fueled by innovations in autoregressive/non-autoregressive (AR/NAR) models [4, 20, 25, 34] and diffusion frameworks [5, 18, 30, 35, 36]. Current methods, exemplified by StableMofusion [18] and Momask [11], can generate highly realistic motions closely matching input text prompts. Despite these advancements, a critical evaluation issue persists: standard quantitative metrics often yield scores near or exceeding ground truth (GT) levels [3, 11, 18]. **But do these high scores truly reflect a generation quality surpassing the original ground truth?** Figure 1 presents the generation visualizations of several models that outperform the ground truth (GT). It is evident that, compared to real motion capture data, current models still lack fine-grained detail and often fail to faithfully align with the intended semantics. This growing disconnect highlights the inadequacy of current evaluation practices. Reliance on metrics

like FID [17] and R-Precision [12] is insufficient as they may not effectively capture motion complexities within feature embeddings. However, much contemporary research [11, 18, 30, 34] continues to prioritize optimizing these potentially flawed indicators. This focus risks steering the text-to-motion field towards optimizing surrogate objectives rather than genuine perceptual quality, thereby hindering substantive progress. The development of reliable, efficient, and perceptually grounded evaluation metrics is therefore not just desirable but imperative for the healthy advancement of this domain.

Evaluating synthesized motion presents unique challenges due to its high-dimensional, temporal, and continuous characteristics [13]. Unlike image generation [33], where attribute-focused and human preference metrics are prevalent, motion evaluation cannot easily adopt similar paradigms. Attempts to circumvent this complexity by decomposing motion into discrete or linguistic units [11, 34] have shown limited generalization across diverse datasets. Consequently, the field predominantly relies on metrics calculated from features extracted by a foundational text-motion encoder [12, 14]. However, the representational power of this encoder is increasingly strained by the sophistication of modern synthesis models [21, 32, 37]. This limitation directly impacts the utility of the most common metrics: FID compares distributions of motion embeddings and R-Precision measures semantic alignment. Both metrics are limited by the shortcomings of the text-motion encoder, hindering their reliability and effectiveness in assessing the quality and fidelity of state-of-the-art text-to-motion synthesis.

R-Precision exhibits two significant drawbacks in evaluating text-to-motion generation. Firstly, as mentioned above, its effectiveness is hampered by the limitations of underlying motion-text embedding models. This is evident in the low ground-truth performance, exemplified by a mere 0.511 Top-1 accuracy [14] on the HumanML3D dataset [12]. To address this specific embedding limitation, we introduce MoCLIP. Fine-tuned via a two-stage strategy on diverse motion datasets, MoCLIP achieves a substantially higher Top-1 accuracy of 0.679. Secondly, R-Precision is undermined by its handling of highly similar text descriptions prevalent in datasets (e.g., "A person walks forward" vs. "A person is walking forward"), which challenge robust encoder differentiation. The metric's reliance on simple ranking within a local subset (like a batch) focuses primarily on whether any correct-seeming text achieves a high rank. If multiple near-duplicates exist, R-Precision can significantly misjudge the results even if the exact ground truth text ranks slightly lower (e.g., 4th) among its close variants. This local "high-score seeking" behavior inflates the results, as models only need to generate diverse motions that are favored by the embedding model to obtain high scores. This mechanism can inadvertently penalize GT slightly while over-rewarding generated samples that fit the encoder's learned preferences, highlighting the need for a more globally comprehensive assessment beyond local top ranks [18].

The Fréchet Inception Distance (FID) [17] is fundamentally constrained by its Gaussian distribution assumption for both real and generated data. This assumption fails to hold for motion datasets (Figure 4) [19]. Furthermore, FID's estimation process, which relies on finite samples to compute moments and utilizes potentially limited feature representations, introduces bias. This bias contributes

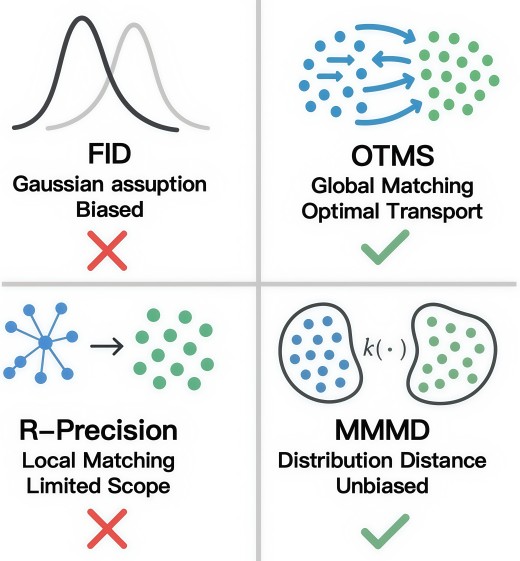

**Figure 2: Conceptual Differences in Evaluation Metrics. Current text-to-motion evaluation often relies on FID, which assumes Gaussian distributions and is biased, or R-Precision, based on limited local matching. These limitations motivate the exploration of alternative metrics like OTMS, which employs global optimal transport, and MMMD, offering an unbiased distribution distance, to achieve more reliable assessments.**

significantly to the divergence between FID scores and human perceptual assessments of motion realism [19]. FID is also known for its inefficiency and instability, displaying considerable sensitivity to evaluation sample size [5]. The biggest concern lies in its potential for paradoxical evaluation: FID scores might decrease as motion quality improves initially but can subsequently increase once quality surpasses an indeterminate point, rendering the metric unreliable. Such unreliability becomes especially acute when evaluating high-performing modern models achieving very low FID scores, precisely where the metric's biased nature has the most severe consequences.

Inspired by Optimal Transport [24] we redefine the R-Precision ranking problem as a matching problem and introduce OTMS (Optimal Transport Matching Score), a global evaluation metric specifically tailored to address existing method limitations. Building on MoCLIP embeddings for both textual and motion data, we first construct a cost matrix from their cosine similarities and then apply a regularized optimal transport algorithm (i.e., the Sinkhorn method [8]) to derive a single cost value that reflects the overall alignment quality between texts and motions. An intuitive way to understand optimal transport is through the classic "moving sand" analogy: each batch of text embeddings can be viewed as piles of sand, and each batch of motion embeddings as the holes that need to be filled. The optimal transport plan determines how best to globally "distribute" the sand (i.e., text representation) into holes (i.e.,

motion representation) with minimal total cost—thereby transcending simpler, local matching strategies like direct cosine similarity. Empirically, we observe that this global perspective yields a more faithful reflection of text-to-motion correspondence compared to conventional retrieval-based metrics.

Concurrently, prior work [19] demonstrated that the CLIP Embeddings Maximum Mean Discrepancy Distance (CMMD) serves as a superior alternative to FID. Inspired by this finding, we adapt this methodology to the text-to-motion (T2M) domain, introducing the MoCLIP Embeddings Maximum Mean Discrepancy Distance (MMMD) metric. MMMD encodes motions with MoCLIP, computes an unbiased distributional divergence using a characteristic kernel, and crucially avoids any Gaussian assumptions often required by the Fréchet distance. Our experiments confirm that MMMD is more robust to variations in sample size and supports efficient, parallelizable computation, making it especially well-suited for real-world scenarios where computational overhead is a concern.

In this paper, we rigorously validate the proposed Optimal Transport Matching Score (OTMS) and MoCLIP-based Maximum Mean Discrepancy (MMMD) metrics. Through extensive experiments and human evaluations, we demonstrate their effectiveness in overcoming the limitations of traditional approaches. Our results confirm that OTMS and MMMD provide assessments that align more closely with human perceptual judgments, offering a more reliable, efficient, and comprehensive evaluation framework. This work establishes a more robust foundation for measuring and guiding future advancements in text-to-motion synthesis. We summarize our contributions as follows:

(1) We are the first to systematically highlight the limitations of current evaluation metrics in text-to-motion generation, analyzing their shortcomings from multiple perspectives.

(2) We propose two new, more reliable metrics—OTSM and MMMD which are built upon MoCLIP, a motion-text embedding model with stronger encoding precision and semantic representation capability, enabling more robust and faithful evaluation of text-to-motion generation.

(3) Extensive experiments demonstrate that OTSM and MMMD better reflect model performance and align more consistently with human perception compared to existing metrics.

## 2 RELATED WORK

**Text-to-Motion Generation.** The field of text-to-motion (T2M) generation has significantly advanced, largely spurred by the creation of large-scale datasets such as HumanML3D [12], KIT-ML [28], and CombatMotion [32]. Initial research efforts explored foundational techniques, including sequence-to-sequence models [1] and early vector-quantization frameworks [15]. While these methods provided valuable insights into mapping language to motion, they often struggled to capture the intricate complexities and diversity inherent in human movement. Subsequent research introduced more sophisticated architectures to enhance generation quality. Diffusion models [6, 10, 29, 31, 35, 36] leveraged iterative denoising for high-fidelity and varied outputs. Variational Autoencoder (VAE) based approaches like TEMOS [23] and TEACH [2] improved the modeling of complex motion distributions. Furthermore, Transformer-based

models such as MotionGPT [20] and T2M-GPT [34] have demonstrated strong capabilities in capturing long-range dependencies and contextual understanding.

Recent advancements have pushed the boundaries of T2M, focusing on semantic consistency, stylization, and controllability. State-of-the-art models, including MoMask [11], StableMoFusion [18], and MotionCLR [3], now achieve remarkable performance using advanced latent representations and refinement techniques. Notably, their performance on established evaluation metrics primarily R-Precision, Frechet Inception Distance (FID), and Diversity metrics [12] is exceptionally high, occasionally even surpassing the ground truth recordings according to these scores. However, this success highlights a growing limitation: the inadequacy of current evaluation metrics. These standard metrics often fail to capture critical nuances, such as the fine-grained accuracy of semantic alignment between text and motion, subtle physical plausibility details, or the potential for user interaction. Recognizing this gap, our work introduces two novel evaluation metrics designed to address these shortcomings. We aim to provide a more comprehensive and interpretable assessment framework, contributing to the development of truly robust and high-quality T2M synthesis systems.

## 3 Preliminaries

In this part, we introduce the fundamental concepts of optimal transport (OT) (3.1) and maximum mean discrepancy (MMD) (3.2) in the context of comparing probability distributions. These form the theoretical underpinnings of our proposed evaluation framework.

### 3.1 Optimal Transport

Consider two discrete probability distributions over a metric space, denoted by

$$\mu = \sum_{i=1}^{M} p_i\, \delta_{z_i} \quad \text{and} \quad \nu = \sum_{j=1}^{N} q_j\, \delta_{y_j}, \tag{1}$$

where $\delta_{z_i}$ and $\delta_{y_j}$ are Dirac delta functions at locations $z_i$ and $y_j$, respectively. Let $\mathbf{p} = (p_1, \ldots, p_M)^\top$ and $\mathbf{q} = (q_1, \ldots, q_N)^\top$ be the corresponding probability vectors, satisfying $\sum_{i=1}^{M} p_i = \sum_{j=1}^{N} q_j = 1$. Let $C \in \mathbb{R}^{M \times N}$ denote the pairwise cost matrix, with $C_{ij} = c(z_i, y_j)$, which quantifies the "distance" or "cost" of matching $z_i$ with $y_j$.

To measure the dissimilarity between $\mu$ and $\nu$, the classical OT problem seeks an optimal transport plan $\mathbf{P}^* \in \mathbb{R}^{M \times N}$ that minimizes the total matching cost:

$$\mathbf{P}^* = \underset{\mathbf{P} \in \Gamma(\mathbf{p}, \mathbf{q})}{\arg\min} \sum_{i=1}^{M} \sum_{j=1}^{N} P_{ij}\, C_{ij}, \tag{2}$$

subject to the marginal constraints $\mathbf{P}\,\mathbf{1}_N = \mathbf{p}$ and $\mathbf{P}^\top\,\mathbf{1}_M = \mathbf{q}$, where $\Gamma(\mathbf{p}, \mathbf{q})$ is the set of all feasible transport plans (matrices $\mathbf{P} \geq 0$ satisfying the marginal constraints). In practice, entropy regularization [8] is frequently applied to improve stability and computational efficiency. The regularized problem is formulated as

$$\mathbf{P}^* = \underset{\mathbf{P} \in \Gamma(\mathbf{p}, \mathbf{q})}{\arg\min} \sum_{i,j} P_{ij}\, C_{ij} - \lambda\, H(\mathbf{P}), \tag{3}$$

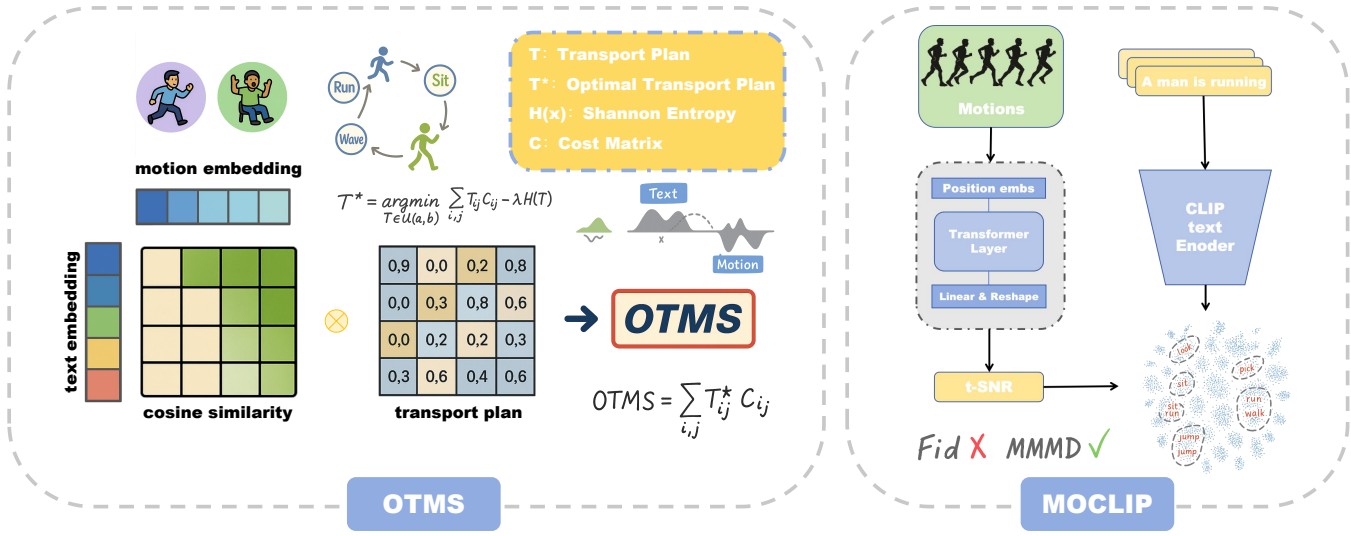

**Figure 3: Illustration of the OTMS and MoCLIP framework. On the left, OTMS utilizes optimal transport to align motion and text embeddings, with the transport plan minimizing the cost (cosine similarity) between the two distributions. On the right, MoCLIP incorporates a transformer-based architecture to align motion and text representations, utilizing CLIP for text encoding and MoCLIP embeddings for semantic alignment, with evaluation metrics such as FID (traditional) and MMMD (proposed) applied using these representations.**

where $\lambda$ is the regularization coefficient, and $H(\mathbf{P}) = -\sum_{i,j} P_{ij} \log P_{ij}$ is the Shannon entropy of $\mathbf{P}$. This relaxed objective can be solved efficiently via the Sinkhorn-Knopp algorithm [9]. Optimal transport, particularly in its entropy-regularized form, has gained prominence for robustly comparing complex distributions by aligning them according to both local and global structures.

### 3.2 Maximum Mean Discrepancy

Maximum Mean Discrepancy (MMD) [19] provides a non-parametric, kernel-based approach to quantify distributional discrepancies without assuming any specific parametric form. Let $\mathbf{X} = \{x_1, \ldots, x_N\}$ and $\mathbf{Y} = \{y_1, \ldots, y_M\}$ be samples drawn i.i.d. from distributions $P$ and $Q$, respectively. MMD is defined with respect to a reproducing kernel Hilbert space (RKHS) $\mathcal{H}$ endowed with a characteristic kernel $\kappa$.

Formally, the squared MMD between $P$ and $Q$ is:

$$\begin{aligned}
\text{MMD}^2(P, Q) = \ &\mathbb{E}_{x, x' \sim P}[\kappa(x, x')] + \mathbb{E}_{y, y' \sim Q}[\kappa(y, y')] \\
&- 2\,\mathbb{E}_{x \sim P,\, y \sim Q}[\kappa(x, y)],
\end{aligned} \quad (4)$$

where $x, x' \in \mathbf{X}$, $y, y' \in \mathbf{Y}$.

For discrete samples, an unbiased empirical estimator of $\text{MMD}^2$ is

$$\begin{aligned}
\widehat{\text{MMD}}^2(\mathbf{X}, \mathbf{Y}) = \ &\frac{1}{N(N-1)} \sum_{i \neq j} \kappa(x_i, x_j) \\
&+ \frac{1}{M(M-1)} \sum_{i \neq j} \kappa(y_i, y_j) \\
&- \frac{2}{NM} \sum_{i=1}^{N} \sum_{j=1}^{M} \kappa(x_i, y_j).
\end{aligned} \quad (5)$$

A commonly used kernel for MMD is the Gaussian Radial Basis Function (RBF) kernel, $\kappa(u, v) = \exp(-\|u - v\|^2/(2\sigma^2))$, with $\sigma$ being a bandwidth parameter. Crucially, if $\kappa$ is characteristic, $\text{MMD}^2(P, Q) = 0$ if and only if $P = Q$. Hence, MMD effectively captures both mean and higher-order discrepancies between distributions, making it suitable for complex, high-dimensional datasets.

Optimal transport and MMD thus offer complementary perspectives on comparing probability distributions. OT aligns sample points explicitly through a transport plan, while MMD leverages kernel embeddings to compare distributions in a Hilbert space. In the subsequent sections, we will utilize both approaches to rigorously assess the semantic alignment of motion data within text-to-motion generation tasks.

## 4 Evaluation Metric

In this section we analyze the limitations of traditional evaluation metrics such as FID and Top-K and address these shortcomings by introducing two novel metrics: OTMS based on Optimal Transport and MMD based on Maximum Mean Discrepancy.

### 4.1 Limitations of Traditional Evaluation Metrics

The conventional use of Fréchet Inception Distance (FID) in motion evaluation inherits intrinsic limitations from its image analysis origins while introducing additional motion-specific vulnerabilities. Initial implementations employed motion encoders, such as the autoencoder proposed by Guo et al. [14], which demonstrated constrained representational power, achieving only 0.797 in top-3 action recognition accuracy on the HumanML3D dataset [12]. Despite improvements in motion encoders, fundamental issues persist due to the intrinsic formulation of the FID metric:

$$\text{FID}(P, Q) = \|\mu_P - \mu_Q\|_2^2 + \text{Tr}\left(\Sigma_P + \Sigma_Q - 2(\Sigma_P \Sigma_Q)^{1/2}\right). \quad (6)$$

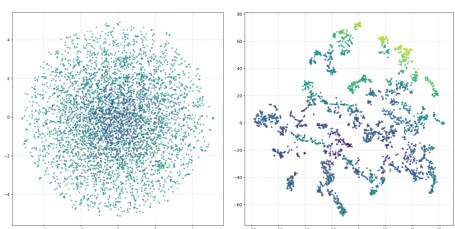

**Figure 4: The left panel displays simulation results of the standard normal distribution, while the right panel visualizes the motion embedding distribution generated via t-SNE dimensionality reduction. Notably, the low-dimensional manifold of motion embeddings deviates significantly from Gaussian characteristics, indicating the presence of complex nonlinear structures inherent in the original high-dimensional motion feature space, which fundamentally contradicts Gaussian assumptions.**

FID inherently relies on two problematic assumptions in the context of motion analysis. Firstly, it presupposes multivariate normality, directly conflicting with the hierarchical, spatiotemporal complexity of human motion. In figure 4, empirical evidence from t-SNE visualizations of 10,000 samples from the HumanML3D test dataset [12] clearly reveals distinct, multi-modal clusters corresponding to action semantics, such as periodic locomotion versus discrete gestures. Moreover, Mardia's multivariate normality test emphatically rejects Gaussianity, yielding extreme skewness ($\chi^2 = 1.86 \times 10^{12}$, df = 22,500,864) and kurtosis statistics ($z = 11,848.6$), both with $p$-values lower than $10^{-324}$ ($d = 512$, $n = 29,184$).

Secondly, and crucially, FID is a **biased estimator** of the true Fréchet distance. This bias arises due to the nonlinear operation involved in estimating covariance matrices from finite samples, specifically the matrix square root term $(\Sigma_r \Sigma_g)^{1/2}$. Under limited sample conditions, this nonlinearity systematically results in underestimation of this term, consequently causing an *overestimation* of the FID score. Such theoretical deficiencies persist irrespective of the encoder's quality, further evidenced by controlled experiments where state-of-the-art motion generators paradoxically improved FID scores despite qualitative degradation observed in human studies.

These limitations underscore the necessity for distribution-free evaluation metrics that more reliably reflect semantic coherence in generated motions, thus motivating our proposed metrics detailed in Section 4.2.

Top-$k$ metrics, notably R-Precision, are widely used for text-to-motion evaluation due to their intuitive interpretation and ease of computation. These metrics assess the capability of retrieving correct motions from a predefined candidate set, typically containing 32 items. Despite their popularity, Top-$k$ metrics primarily evaluate local retrieval accuracy and fail to comprehensively capture the semantic alignment between motion and textual embeddings.

A critical drawback arises from the limited retrieval accuracy observed even with ground-truth motions. Empirical human evaluations indicate that ground-truth retrieval performance exhibits a natural ceiling (approximately 0.9 for Top-3 accuracy), highlighting

inherent dataset misalignments. Consequently, relying exclusively on Top-$k$ metrics risks misrepresenting the true semantic alignment capability of state-of-the-art models, making performance improvements difficult to interpret accurately.

Additionally, recent models surpassing ground-truth Top-$k$ scores exacerbate interpretability concerns. Improvements in retrieval metrics may result from biases in the embedding space or optimization procedures, rather than genuine advancements in semantic understanding. Moreover, Top-$k$ metrics neglect the broader distributional structure within embedding spaces, failing to capture continuous semantic alignment comprehensively. These limitations necessitate a more robust and holistic evaluation framework beyond local retrieval accuracies.

## 4.2 Proposed Evaluation Metrics

To overcome the aforementioned limitations associated with traditional metrics, we introduce *MoCLIP*, accompanied by two novel evaluation metrics specifically designed for text-to-motion tasks: *Optimal Transport Matching Score (OTMS)* and *MoCLIP-based Maximum Mean Discrepancy (MMMD)*. Inspired by recent advancements in text-to-image evaluations [19], these metrics leverage MoCLIP's enhanced semantic alignment capabilities.

*4.2.1 Optimal Transport Matching Score (OTMS).* We define the Optimal Transport Motion Semantic (OTMS) metric, which is showed in figure 5, based on normalized embeddings of motions $\{\mathbf{m}_i\}_{i=1}^{M}$ and texts $\{\mathbf{t}_j\}_{j=1}^{N}$ extracted by MoCLIP. In contrast to traditional retrieval-based metrics, OTMS leverages the global semantic alignment between text and motion distributions through an optimal transport formulation.

Specifically, we first construct a cost matrix $\mathbf{C} \in \mathbb{R}^{M \times N}$ utilizing cosine similarity to quantify pairwise semantic distances:

$$C_{ij} = 1 - \langle \mathbf{m}_i, \mathbf{t}_j \rangle. \tag{7}$$

We then define discrete uniform probability distributions, represented by vectors $\mathbf{a} = \frac{1}{M}\mathbf{1}_M$ and $\mathbf{b} = \frac{1}{N}\mathbf{1}_N$, over the motion and text embeddings, respectively, and employ the Sinkhorn algorithm [**?** ] to obtain the optimal transport plan $\mathbf{T}^*$:

$$\mathbf{T}^* = \arg \min_{\mathbf{T} \in \Gamma(\mathbf{a}, \mathbf{b})} \sum_{i=1}^{M} \sum_{j=1}^{N} T_{ij} C_{ij} - \lambda H(\mathbf{T}), \tag{8}$$

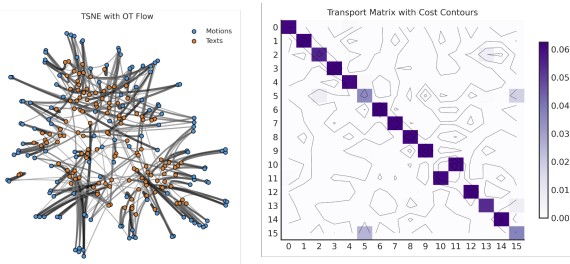

**Figure 5: The left figure shows the distribution matching situation after the MoCLIP encodings of 128 motions and texts are reduced in dimension using t-SNE. The right figure is the heatmap of the transport plan.**

where $H(\cdot)$ denotes the entropy regularization term, $\lambda$ is the corresponding regularization coefficient, and $\Gamma(\mathbf{a}, \mathbf{b})$ is the set of transport plans matching the uniform marginals $\mathbf{a}$ and $\mathbf{b}$. Finally, the OTMS metric is computed as:

$$\text{OTMS} = \sum_{i=1}^{M} \sum_{j=1}^{N} T_{ij}^* C_{ij}, \tag{9}$$

which is essentially the Sinkhorn distance between the two embedding distributions. Lower OTMS values indicate stronger global semantic alignment.

*4.2.2 MoCLIP-Based Maximum Mean Discrepancy (MMMD).* To measure distributional divergence without assuming Gaussianity, we propose the MoCLIP-based Maximum Mean Discrepancy (MMMD). Let $P$ denote the distribution of MoCLIP embeddings from ground-truth motions $\{\mathbf{m}_i\}$ and $Q$ denote the distribution from generated sequences $\{\hat{\mathbf{m}}_j\}$. The squared MMD employs a characteristic kernel $\kappa$, such as the Gaussian RBF, and is defined via expectations:

$$\text{MMD}^2(P, Q) = \mathbb{E}_{\mathbf{m},\mathbf{m}'\sim P}[\kappa(\mathbf{m}, \mathbf{m}')] + \mathbb{E}_{\hat{\mathbf{m}},\hat{\mathbf{m}}'\sim Q}[\kappa(\hat{\mathbf{m}}, \hat{\mathbf{m}}')] \\ - 2\mathbb{E}_{\mathbf{m}\sim P,\hat{\mathbf{m}}\sim Q}[\kappa(\mathbf{m}, \hat{\mathbf{m}})], \tag{10}$$

where expectations are taken over independent samples drawn from $P$ and $Q$.

In practice, given finite sets of MoCLIP embeddings $\mathbf{X} = \{\mathbf{m}_1, \ldots, \mathbf{m}_N\}$ drawn from $P$ and $\mathbf{Y} = \{\hat{\mathbf{m}}_1, \ldots, \hat{\mathbf{m}}_M\}$ drawn from $Q$, we compute the unbiased empirical estimate of $\text{MMD}^2(P, Q)$ as:

$$\widehat{\text{MMD}}^2(\mathbf{X}, \mathbf{Y}) = \frac{1}{N(N-1)} \sum_{i=1}^{N} \sum_{j\neq i} \kappa(\mathbf{m}_i, \mathbf{m}_j)$$
$$+ \frac{1}{M(M-1)} \sum_{i=1}^{M} \sum_{j\neq i} \kappa(\hat{\mathbf{m}}_i, \hat{\mathbf{m}}_j) \tag{11}$$
$$- \frac{2}{NM} \sum_{i=1}^{N} \sum_{j=1}^{M} \kappa(\mathbf{m}_i, \hat{\mathbf{m}}_j).$$

and define MMMD as:

$$\text{MMMD} = \alpha \sqrt{\widehat{\text{MMD}}^2(\mathbf{X}, \mathbf{Y})}, \tag{12}$$

with scaling factor $\alpha = 1000$ to improve readability.

Unlike FID, the nonparametric formulation of MMMD offers an unbiased estimator of distributional discrepancy, inherently accommodating the complex, non-Gaussian, and high-dimensional nature of motion data. By directly comparing empirical distributions without relying on parametric assumptions, MMMD yields a semantically coherent measure that correlates strongly with human perceptual judgments, offering a more principled and reliable evaluation of generative quality.

## 5 Experiments

Our experimental evaluation utilizes the HumanML3D [12] and KIT-ML [28] datasets. The HumanML3D dataset contains 14,616 motions sourced from AMASS [22] and HumanAct12 [16]. Each motion is paired with three textual descriptions, resulting in 44,970

descriptions total. This dataset encompasses diverse actions, including walking, exercising, and dancing. The KIT-ML dataset provides 3,911 motions and 6,278 corresponding text descriptions. We assess model performance using our proposed evaluation metrics: Optimal Transport Matching Score (OTMS) and MoCLIP-based Maximum Mean Discrepancy (MMMD).

### 5.1 Experiment Setting

We compare several state-of-the-art text-to-motion generation models, including StableMoFusion, MDM, T2MT, MoMask, MMM, BAMM, T2M-GPT, and Discord. For evaluation, the proposed metrics OTMS, MMMD, and MoCLIP-based R-Precision were computed using embeddings extracted from a pretrained MoCLIP encoder, which provides a shared semantic space across modalities. In contrast, the metrics FID and the original R-Precision were calculated using embeddings from their respective original encoders. All experiments were conducted on an NVIDIA RTX 4090 GPU. Reported metrics represent the average over 20 independent runs, presented with 95% confidence intervals for robust evaluation.

### 5.2 Effectiveness of New Metrics

Table 1 and Table 2 present the quantitative results obtained from evaluating multiple state-of-the-art models on the HumanML3D and KIT-ML datasets, respectively. For ensuring the reliability of comparisons, each experiment was conducted 20 times, with results reported along with 95% confidence intervals. Our experiments involved a variety of models ranging from earlier proposed frameworks, such as T2MT, to more recent models like StableMoFusion. Results demonstrate that earlier models consistently exhibit poorer performance on our metrics (OTMS and MDM), while newer approaches generally perform better. This trend indicates that our metrics effectively distinguish between higher- and lower-quality models. Additionally, OTMS addresses the common problem observed in prior metrics, where some models incorrectly outperform ground-truth (GT) data. Nevertheless, to further validate the robustness and accuracy of our proposed metrics, human evaluation studies are conducted, as detailed in the next subsection.

### 5.3 Human Evaluation

To assess whether our metrics align with human intuition, we conducted a human evaluation study employing 16 evaluators. Each participant assessed a total of 4000 samples, consisting of 1000 samples each from MDM, StableMoFusion, MoMask, and ground truth (GT). Evaluations were performed across four core dimensions: motion completeness, directional and angular accuracy, appropriate utilization of body parts, and physical plausibility. These dimensions were chosen to comprehensively capture the fidelity of motion-to-text matching, with each dimension independently assessed yet collectively contributing to the overall evaluation.

Results from the human evaluation closely align with our proposed metrics, specifically OTMS and MMD, highlighting their strong correlation with human judgment. Notably, we observed that in cases where the FID score is exceptionally low, our MMD metric more accurately reflects human preferences. Leveraging these insights, we constructed a new human-preference dataset that can serve as a robust benchmark for future evaluations of

| Method | FID ↓ | MoCLIP R-Precision ↑ | | | R-Precision ↑ | | | OTMS ↓ | MMMD ↓ |
|---|---|---|---|---|---|---|---|---|---|
| | | TOP1 | TOP2 | TOP3 | TOP1 | TOP2 | TOP3 | | |
| GT | $0.002^{\pm0.000}$ | $0.679^{\pm0.002}$ | $0.836^{\pm0.003}$ | $0.896^{\pm0.002}$ | $0.511^{\pm0.003}$ | $0.703^{\pm0.003}$ | $0.797^{\pm0.002}$ | $0.481^{\pm0.003}$ | $0.003^{\pm0.001}$ |
| MoMask [11] | $0.045^{\pm0.002}$ | $0.679^{\pm0.002}$ | $0.836^{\pm0.003}$ | $0.896^{\pm0.002}$ | $0.521^{\pm0.002}$ | $0.713^{\pm0.002}$ | $0.807^{\pm0.002}$ | $0.495^{\pm0.001}$ | $0.190^{\pm0.001}$ |
| Discord [7] | $0.032^{\pm0.002}$ | $0.687^{\pm0.002}$ | $0.842^{\pm0.002}$ | $0.902^{\pm0.002}$ | $0.524^{\pm0.003}$ | $0.715^{\pm0.003}$ | $0.809^{\pm0.002}$ | $0.489^{\pm0.0010}$ | $0.178^{\pm0.002}$ |
| MMM [27] | $0.080^{\pm0.003}$ | $0.651^{\pm0.002}$ | $0.806^{\pm0.002}$ | $0.870^{\pm0.002}$ | $0.504^{\pm0.003}$ | $0.696^{\pm0.003}$ | $0.794^{\pm0.002}$ | $0.488^{\pm0.003}$ | $0.055^{\pm0.001}$ |
| BAMM [26] | $0.055^{\pm0.002}$ | $0.662^{\pm0.002}$ | $0.822^{\pm0.002}$ | $0.888^{\pm0.002}$ | $0.525^{\pm0.002}$ | $0.720^{\pm0.003}$ | $0.814^{\pm0.003}$ | $0.499^{\pm0.0001}$ | $0.220^{\pm0.003}$ |
| T2M-GPT [34] | $0.492^{\pm0.003}$ | $0.630^{\pm0.003}$ | $0.791^{\pm0.003}$ | $0.861^{\pm0.002}$ | $0.492^{\pm0.003}$ | $0.679^{\pm0.002}$ | $0.775^{\pm0.002}$ | $0.487^{\pm0.001}$ | $0.033^{\pm0.013}$ |
| StableMofusion [18] | $0.098^{\pm0.003}$ | $0.735^{\pm0.002}$ | $0.876^{\pm0.002}$ | $0.926^{\pm0.002}$ | $0.553^{\pm0.003}$ | $0.748^{\pm0.002}$ | $0.841^{\pm0.002}$ | $0.478^{\pm0.006}$ | $0.025^{\pm0.002}$ |
| MDM [30] | $0.544^{\pm0.044}$ | $0.611^{\pm0.006}$ | $0.781^{\pm0.004}$ | $0.855^{\pm0.004}$ | $0.455^{\pm0.006}$ | $0.465^{\pm0.007}$ | $0.749^{\pm0.006}$ | $0.506^{\pm0.001}$ | $0.555^{\pm0.0012}$ |
| T2MT [15] | $1.501^{\pm0.017}$ | $0.509^{\pm0.002}$ | $0.679^{0.002}$ | $0.764^{\pm0.000}$ | $0.424^{\pm0.003}$ | $0.618^{\pm0.003}$ | $0.729^{\pm0.002}$ | $0.524^{\pm0.003}$ | $0.592^{\pm0.001}$ |

Table 1: Quantitative comparison of various text-to-motion generation methods on the HumanML3D dataset using multiple evaluation metrics. We report FID (Fréchet Inception Distance; lower is better), MoCLIP-based R-Precision (Top-1/2/3; higher is better), traditional CLIP-based R-Precision, OTMS (Optimal Transport Matching Score; lower is better), and MMMD (MoCLIP-based Maximum Mean Discrepancy; lower is better). The results demonstrate the effectiveness of MoCLIP-based metrics in better distinguishing semantic alignment and distribution consistency. GT denotes ground truth motion. All metrics are averaged 20 runs with 95% confidence intervals.

| Method | FID ↓ | MoCLIP R-Precision ↑ | | | R-Precision ↑ | | | OTMS | CMMD |
|---|---|---|---|---|---|---|---|---|---|
| | | TOP1 | TOP2 | TOP3 | TOP1 | TOP2 | TOP3 | | |
| GT | $0.031^{\pm0.004}$ | $0.556^{\pm0.005}$ | $0.759^{\pm0.006}$ | $0.860^{\pm0.005}$ | $0.424^{\pm0.005}$ | $0.649^{\pm0.006}$ | $0.779^{\pm0.006}$ | $0.681^{\pm0.001}$ | $-0.016^{\pm0.000}$ |
| Momask [11] | $0.204^{\pm0.110}$ | $0.399^{\pm0.007}$ | $0.597^{\pm0.005}$ | $0.714^{\pm0.005}$ | $0.433^{\pm0.007}$ | $0.656^{\pm0.005}$ | $0.781^{\pm0.005}$ | $0.715^{\pm0.000}$ | $2.627^{\pm0.033}$ |
| T2M-GPT [34] | $0.514^{\pm0.029}$ | $0.367^{\pm0.008}$ | $0.566^{\pm0.009}$ | $0.680^{\pm0.009}$ | $0.416^{\pm0.060}$ | $0.627^{\pm0.006}$ | $0.745^{\pm0.006}$ | $0.730^{\pm0.002}$ | $0.175^{\pm0.000}$ |
| StableMofusion [18] | $0.258^{\pm0.029}$ | $0.336^{\pm0.005}$ | $0.518^{\pm0.006}$ | $0.636^{\pm0.005}$ | $0.445^{\pm0.006}$ | $0.660^{\pm0.005}$ | $0.782^{\pm0.004}$ | $0.763^{\pm0.001}$ | $0.204^{\pm0.001}$ |
| MDM [30] | $0.547^{\pm0.069}$ | $0.303^{\pm0.004}$ | $0.613^{\pm0.005}$ | $0.487^{\pm0.000}$ | $0.404^{\pm0.019}$ | $0.615^{\pm0.013}$ | $0.737^{\pm0.005}$ | $0.782^{\pm0.001}$ | $0.793^{\pm0.000}$ |
| T2MT [15] | $0.360^{\pm0.153}$ | $0.223^{\pm0.003}$ | $0.358^{\pm0.005}$ | $0.449^{\pm0.006}$ | $0.280^{\pm0.005}$ | $0.463^{\pm0.006}$ | $0.587^{\pm0.005}$ | $0.803^{\pm0.001}$ | $0.786^{\pm0.016}$ |

Table 2: Quantitative comparison of various text-to-motion generation methods on the KIT-ML dataset using multiple evaluation metrics. We report FID (Fréchet Inception Distance; lower is better), MoCLIP-based R-Precision (Top-1/2/3; higher is better), traditional CLIP-based R-Precision, OTMS (Optimal Transport Matching Score; lower is better), and MMMD (MoCLIP-based Maximum Mean Discrepancy; lower is better). The results demonstrate the effectiveness of MoCLIP-based metrics in better distinguishing semantic alignment and distribution consistency. GT denotes ground truth motion. All metrics are averaged 20 runs with 95% confidence intervals.

motion-text alignment methods. The human evaluation results are presented in Table 3. We also calculated the correlation between human scores and metrics such as FID, OTMS, MMD, and TOPK, as shown in Figure 6. Our proposed metrics rank first and second in this analysis. This indicates that our metrics align better with human preferences.

| model | Score ↑ | FID ↓ | Top3 | OTMS ↓ | MMMD ↓ |
|---|---|---|---|---|---|
| GT | 19.68 | 0.002 | 0.836 | 0.481 | 0.003 |
| MDM | 18.22 | 0.544 | 0.749 | 0.506 | 0.555 |
| StableMofusion | 18.93 | 0.098 | 0.841 | 0.478 | 0.025 |
| Momask | 18.58 | 0.045 | 0.807 | 0.495 | 0.190 |

Table 3: Human Evaluation Results: A comparison of GT, MDM, stablemofusion, and Momask was conducted on 1000 test set samples. Human evaluation scores were given, with a maximum possible score of 20 points.

## 5.4 Disscussion

*5.4.1 Influence of $\lambda$ on OTMS..* The regularization parameter $\lambda$ in the entropy-regularized Sinkhorn algorithm directly impacts computational efficiency and convergence stability. Specifically, smaller $\lambda$ values (e.g., $\lambda < 0.01$) result in sharply peaked transport plans, potentially enhancing local alignment sensitivity but simultaneously increasing the risk of numerical instability and slower convergence. Conversely, larger values of $\lambda$ produce smoother transport plans that may diminish the metric's ability to discriminate fine-grained semantic differences. After systematically evaluating a range of values $\lambda \in \{0.01, 0.02, 0.03, 0.04, 0.05, 0.06, 0.07, 0.08, 0.09, 1.00\}$ ,we found that when computed with a batch size of 32, $\lambda = 0.02$ can achieve the largest ot based Top-k A.4 value on the test set data and also has good efficiency.

*5.4.2 Causal Analysis of the Model's Higher R-Precision Scores over GT.* Our experimental results show that the model-generated motions frequently achieve higher R-Precision and CLIP scores compared to the ground truth (GT), indicating better embedding-level alignment with textual descriptions. However, human evaluations

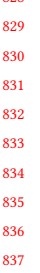
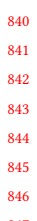
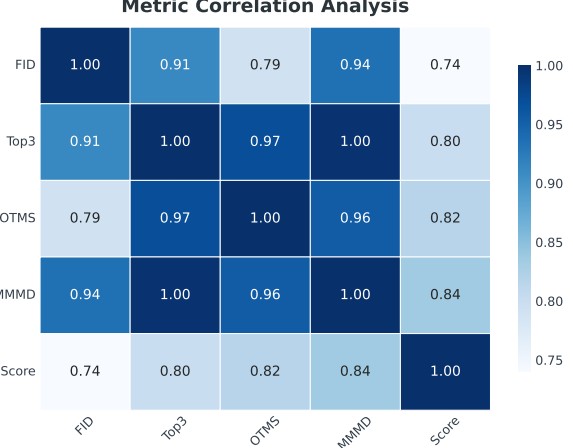

**Comprehensive Model Evaluation Analysis**

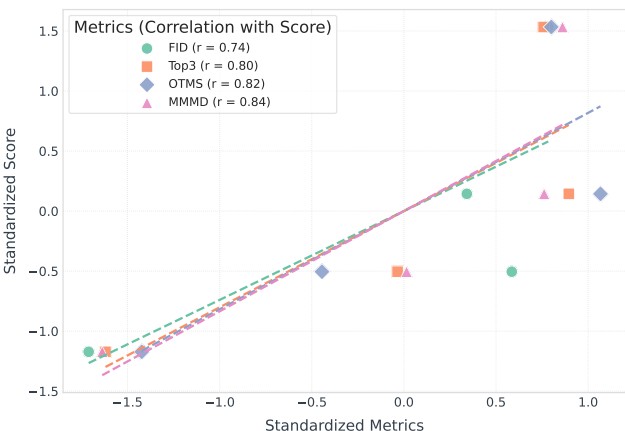

Figure 6: The first figure presents the correlation matrix between our proposed metrics, FID Top-K, and human evaluation scores. The second figure illustrates the correlation analysis between our metrics, FID Top-K, and human evaluation.

consistently suggest that the generated motions are inferior in actual quality compared to GT. We attribute this discrepancy primarily to embedding biases introduced during the training of the motion and text encoders. Specifically, both encoders are trained on a relatively limited dataset, causing them to favor motion patterns similar to those observed during training, thus artificially inflating embedding-based metrics.

To validate this hypothesis, we conducted experiments on the training set itself and found that the GT's top-3 R-Precision score (0.876) outperformed that of StableMofusion (0.843). This indicates that when evaluated within the original data distribution, embedding-based metrics correctly rank GT higher, whereas the same encoders struggle to accurately rank novel motions on limited datasets. This underscores the critical issue of embedding bias

| $\lambda$ | Top1 | Top2 | Top3 | Time (s) |
|------|------|------|------|------|
| 0.01 | 0.782 | 0.900 | 0.940 | 11.58 |
| **0.02** | **0.810** | **0.930** | **0.960** | 9.50 |
| 0.03 | 0.790 | 0.914 | 0.953 | 11.58 |
| 0.04 | 0.793 | 0.914 | 0.951 | 9.90 |
| 0.05 | 0.781 | 0.913 | 0.955 | 7.22 |
| 0.06 | 0.789 | 0.905 | 0.942 | 3.33 |
| 0.07 | 0.772 | 0.898 | 0.944 | 1.77 |
| 0.08 | 0.776 | 0.899 | 0.939 | 1.58 |
| 0.09 | 0.778 | 0.899 | 0.941 | 1.11 |
| 1.00 | 0.775 | 0.898 | 0.942 | 1.05 |

Table 4: Effect of regularization parameter $\lambda$ on OT-based Top-k and inference time. The batch size is set to 32. The best performance is observed at $\lambda = 0.02$.

resulting from insufficient data diversity and scale, highlighting the necessity of larger and more diverse datasets for reliable embedding-based evaluations.

## 6 Conclusion

Traditional text-to-motion metrics like FID and R-Precision often fail evaluations. They misalign with human perception exhibit Gaussian bias focus locally and depend heavily on encoders. We proposed two novel metrics Optimal Transport Matching Score (OTMS) and MoCLIP-based Maximum Mean Discrepancy (MMMD) to address these shortcomings. OTMS leverages optimal transport for global semantic alignment surpassing R-Precision's local matching limitations. MMMD utilizes an enhanced MoCLIP encoder and MMD with RBF kernels providing an unbiased distributional comparison free from FID's Gaussian assumptions and inefficiency. Extensive experiments demonstrate OTMS and MMMD better distinguish model performance correlate strongly with human judgment and avoid the overestimation issues plaguing older metrics. Our work offers a robust efficient and perceptually faithful evaluation framework grounded in global alignment and distribution-free statistics. It highlights existing metric deficiencies and establishes a foundation for reliable assessment crucial for advancing genuine motion quality and semantic fidelity in text-to-motion generation.

**In the future** significant progress in text-to-motion synthesis requires moving beyond improved evaluation metrics. While crucial our proposed OTMS and MMMD address assessment limitations yet fundamental data and representation challenges persist hindering substantial breakthroughs. Current datasets such as HumanML3D and KIT-ML are insufficient necessitating the creation of much larger high idelity datasets. These next generation datasets should feature diverse fine grained textual descriptions suitable perhaps for pretraining scale models. Simultaneously research must explore more expressive motion representations potentially focusing on controllable attributes beyond raw kinematics. Developing methods to learn and rigorously evaluate these attribute based motion models constitutes another crucial research avenue. Addressing interconnected challenges of diverse data, innovative representations and effective evaluation is crucial for achieving high quality semantically faithful text to motion generation.

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

# A   APPENDIX

## A.1   MoCLIP

Inspired by CLIP's image-text alignment success MoCLIP extends this concept to human motion. It learns a shared embedding space mapping textual descriptions to corresponding motion sequences enabling cross-modal retrieval and understanding.

*A.1.1   Architecture.* MoCLIP employs a dual-encoder structure adapting the pretrained CLIP architecture. The text pathway uses a fine-tuned CLIP text encoder generating semantic embeddings $\mathbf{e}_t \in \mathbb{R}^d$. The motion pathway features a dedicated MotionEncoder processing sequences $\mathbf{M} \in \mathbb{R}^{T \times D}$. This encoder applies linear projection $\mathbf{W}_p$ adds sinusoidal positional encoding $\mathbf{P}$ passes the result through $L$ transformer layers handling variable lengths via mask $\mathbf{M}_{mask}$ performs temporal average pooling and finally projects features via $\mathbf{W}_o$ to obtain motion embeddings $\mathbf{e}_m \in \mathbb{R}^d$. The core operation within the transformer layers is multi-head self-attention:

$$\mathbf{h}_l = \text{TransformerLayer}(\mathbf{h}_{l-1}, \mathbf{M}_{mask}). \quad (13)$$

Contrastive learning in the shared $d$-dimensional space aligns these embeddings scaled by temperature $\tau$.

*A.1.2   Loss Function.* MoCLIP utilizes a symmetric contrastive loss to align modalities:

$$\mathcal{L}_{\text{contrastive}} = \frac{1}{2} \left( \mathcal{L}_{\text{motion-to-text}} + \mathcal{L}_{\text{text-to-motion}} \right). \quad (14)$$

This loss averages the motion-to-text and text-to-motion cross-entropy terms. These terms are computed using cosine similarity between L2-normalized motion and text embeddings promoting high similarity for matched pairs and low similarity for mismatched pairs.

*A.1.3   Training Strategy.* Training follows a two-stage strategy for progressive alignment. **Stage 1: Motion Encoder Pretraining.** The CLIP text encoder is frozen. Only the MotionEncoder components are trained optimizing the contrastive loss (Eq. 14). This initially aligns motion features to the fixed text embedding space. **Stage 2: Joint Fine-Tuning.** The final layers of the CLIP text encoder are unfrozen. The entire model is then fine-tuned jointly with a lower learning rate. This allows mutual refinement of both motion and text representations enhancing the joint embedding space. This approach facilitates stable learning and effective cross-modal integration.

## A.2   Performance of MoCLIP

We evaluated MoCLIP's core text-motion alignment capability on the HumanML3D KIT and CMP datasets. Performance was measured using standard Top-$k$ retrieval accuracy (*Top-1 Top-2 Top-3*). Table 5 shows MoCLIP significantly outperforms the baseline across all datasets. Notably on HumanML3D MoCLIP achieves 0.705 Top-1 accuracy versus the baseline's 0.511. On CMP the improvement is also substantial reaching 0.748 Top-1 accuracy compared to 0.335. Consistent gains are observed on the KIT dataset. These results validate MoCLIP's effectiveness in learning accurate text-motion semantic mappings.

|          | Dataset   | Top-1 | Top-2 | Top-3 |
|----------|-----------|-------|-------|-------|
| Baseline | Humanml3d | 0.511 | 0.703 | 0.797 |
| MoCLIP   | Humanml3d | 0.705 | 0.856 | 0.913 |
| Baseline | KIT       | 0.424 | 0.649 | 0.779 |
| MoCLIP   | KIT       | 0.469 | 0.676 | 0.788 |
| Baseline | CMP       | 0.335 | 0.513 | 0.628 |
| MoCLIP   | CMP       | 0.748 | 0.891 | 0.942 |

Table 5: Top-$k$ retrieval accuracy comparison between the baseline and MoCLIP on HumanML3D, KIT, and CMP datasets.

| Method | FID ↓ | R-Precision ↑ | | |
|--------|-------|------|------|------|
| | | **top1** | **top2** | **top3** |
| MDM baseline | $0.544^{\pm 0.044}$ | $0.455^{\pm 0.006}$ | $0.465^{\pm 0.007}$ | $0.749^{\pm 0.006}$ |
| MDM MoCLIP | $0.527^{\pm 0.034}$ | $0.514^{\pm 0.003}$ | $0.719^{\pm 0.001}$ | $0.820^{\pm 0.001}$ |
| Momask baseline | $0.045^{\pm 0.002}$ | $0.521^{\pm 0.002}$ | $0.713^{\pm 0.002}$ | $0.807^{\pm 0.002}$ |
| Momask MoCLIP | $0.065^{\pm 0.002}$ | $0.529,^{\pm 0.002}$ | $0.724^{\pm 0.002}$ | $0.818^{\pm 0.002}$ |
| StableMofusion baseline | $0.098^{\pm 0.003}$ | $0.553^{\pm 0.003}$ | $0.748^{\pm 0.002}$ | $0.841^{\pm 0.002}$ |
| StableMofusion MoCLIP | $0.074^{\pm 0.003}$ | $0.557^{\pm 0.002}$ | $0.753^{\pm 0.001}$ | $0.846^{\pm 0.002}$ |

Table 6: Evaluation results of MoCLIP integration with different models. The table shows the FID (lower is better) and R-Precision (higher is better) at top1, top2, and top3 for MDM, MoMask, and StableMoFusion models with and without MoCLIP. The results demonstrate the positive impact of MoCLIP on improving both FID and R-Precision scores in motion generation tasks.

## A.3   Integrating MoCLIP for Enhanced Generation

To evaluate the practical benefit of MoCLIP's learned representations we integrated its fine-tuned text encoder $\mathcal{T}_{\text{MoCLIP}}$ into existing generation frameworks. Specifically we replaced the native text encoders of StableMoFusion MDM and MoMask with $\mathcal{T}_{\text{MoCLIP}}$. This modification supplies these generators with motion-aligned text embeddings $\mathbf{e}_t^{\text{MoCLIP}}$ leveraging the shared semantic space detailed in Section A.1. The resulting performance improvements detailed in Table 6 demonstrate the efficacy of this approach. Using $\mathcal{T}_{\text{MoCLIP}}$ consistently enhances generation quality across the tested models. This confirms that the MoCLIP encoder effectively extracts motion-relevant semantics transforming text descriptions into representations more conducive to high-fidelity motion synthesis.

## A.4   Top-$k$ Retrieval via Optimal Transport

To measure retrieval performance with Optimal Transport (OT), we first construct a cost matrix $\mathbf{C} \in \mathbb{R}^{n \times n}$ by

$$C_{ij} = 1 - \cos(\mathbf{x}_i, \mathbf{y}_j), \quad (15)$$

where $\mathbf{x}_i$ and $\mathbf{y}_j$ are the normalized embeddings of motion and text, respectively. Let $\mathbf{a}$ and $\mathbf{b}$ be uniform source and target distributions. Given a regularization parameter $\lambda > 0$, we obtain the transport

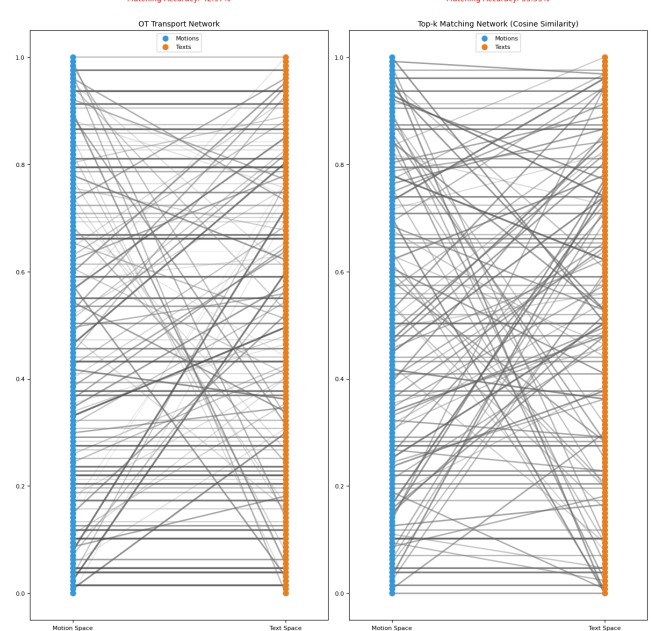

**Figure 7: Comparison of Top-$k$ retrieval using traditional cosine similarity and Optimal Transport (OT) in motion-text alignment. The figure shows the differences between OT-based Top-$k$ retrieval and the traditional Top-$k$ retrieval method. The OT-based method demonstrates higher matching accuracy by allowing more flexible alignments between motion and text pairs. The accuracy values for each method are indicated above the respective plots.**

plan $\mathbf{T}$ by solving

$$\mathbf{T} = \text{Sinkhorn}(\mathbf{a}, \mathbf{b}, \mathbf{C}, \lambda). \tag{16}$$

For Top-$k$ retrieval, each motion sample $i$ ranks text samples based on the row $\mathbf{T}_{i,:}$; those columns $j$ with the largest $T_{ij}$ are deemed the best-aligned text candidates (and vice versa for text-to-motion). This OT-based ranking reflects more flexible matchings than direct cosine similarity alone.

Notably, the OT-based Top-$k$ retrieval method achieves superior matching accuracy compared to traditional Top-$k$ retrieval, as shown in figure 7. This improvement arises because Optimal Transport (OT) allows for a more flexible and nuanced alignment between motion and text embeddings, taking into account the entire distribution of pairwise similarities rather than relying solely on the highest similarity score. The result is a better matching of relevant motion-text pairs, particularly in cases where traditional cosine similarity may fail to capture subtle semantic relationships.

However, different choices of $\lambda$ influence how "peaky" or diffuse $\mathbf{T}$ becomes. A higher $\lambda$ encourages smoother transport, thereby yielding a broader and more distributed alignment, whereas a lower $\lambda$ concentrates on the most salient matches, focusing on sharper alignments between motion and text. In practice, we select the $\lambda$ that maximizes Top-$k$ recall on ground-truth pairs, thereby balancing the trade-off between overly broad and overly rigid alignments.

a4paper, margin=1in

## B  Details of Human Evaluation

For our human evaluation, we selected 1000 motion samples each from MoMask, StableMoFusion, MDM, and the Ground Truth (GT) dataset. We developed custom software specifically designed to facilitate the scoring process by human evaluators. The evaluation focused on assessing the semantic alignment between the generated motion and the input text prompt, using the following questions and scoring scale (detailed below):

---

**Human Evaluation Questions & Scoring**

**[Question 1: Action Completeness]**
*How well does the generated motion include all key action steps described in the text prompt?*
*Score Options: 5 (Complete), 3 (Minor Omission), 0 (Major Omission)*

**[Question 2: Direction/Angle Accuracy]**
*How accurately do the directions and angles in the motion match the text description?*
*Score Options: 5 (Highly Accurate), 3 (Correct Direction, Moderate Angle Deviation), 0 (Incorrect/Severe Error)*

**[Question 3: Body Part Usage]**
*Does the motion utilize the correct body parts as specified in the text, and are they used appropriately?*
*Score Options: 5 (Correct Usage), 3 (Minor Error), 0 (Major Error)*

**[Question 4: Physical Plausibility]**
*How physically plausible and realistic is the generated motion according to physics and human kinematics?*
*Score Options: 5 (Highly Plausible), 3 (Minor Issues), 0 (Severe Issues)*

---

These questions collectively assess the core aspects of text-to-motion generation quality: faithfulness to the prompt's actions (Completeness), spatial precision (Direction/Angle), correct anatomical execution (Body Part Usage), and physical realism (Plausibility). This multi-dimensional approach ensures a comprehensive evaluation of semantic understanding and motion quality.

Below are the detailed scoring guidelines provided to the human evaluators for each question, based on a 3-point scale (5, 3, 0).

