# OpenReview forum: "Towards Better Evaluation Metrics for Text-to-Motion Generation"
_ACM.org/TheWebConf/2026/Workshop/TIME — TIME 2026 Oral_

### Official Review · Reviewer_Q2br · 2025-12-29
**TowardsBetterEvaluationMetricsforText-to-MotionGeneration**

**Rating:** 7
**Confidence:** 3

**Review:**

1) Validate MoCLIP on out-of-distribution data to ensure metrics don't just reflect training data biases.
2) Discuss risks if future generative models directly optimize MoCLIP embeddings to game the metrics.
3) Provide heuristics for selecting the regularization parameter $\lambda$ for datasets of varying sizes.
4) Explicitly detail the Big-O computational complexity of OTMS to assess scalability.
5) Expand human evaluation beyond 16 participants to ensure higher statistical significance.
6) Analyze instances where OTMS/MMMD diverge from human judgment to identify potential blind spots.
7) Investigate why Ground Truth retrieval tops out at ~0.9 to distinguish annotation noise from encoder limitations.
8) Benchmark against a wider variety of recent autoregressive models beyond T2M-GPT.
9) Ablate the specific impact of the two-stage training strategy on the final metric sensitivity.
10) Ensure the release includes exact pre-trained MoCLIP weights to guarantee consistent evaluation across labs.

---

### Official Review · Reviewer_3PUc · 2025-12-29
**The paper has 3 contributions, that obscures which contribution is more effective. Human judgement dataset used for comparison is not public**

**Rating:** 6
**Confidence:** 4

**Review:**

This paper explains the shortcomings of traditional evaluation metrics (FID, R-Precision) in text-to-motion generation (they back this up with Mardia's test as well). The paper proposes MMMD and OTMS metrics, explains how they overcome limitations of existing metrics and it also proposes new MoCLIP embeddings. Validation includes a study with 16 human evaluators assessing 4k samples to demonstrate proposed metrics align more with human judgement. Comments:
- IMHO Einstein quote should be removed from Introduction
- No references found to Figures 1,2,3,5,7; Table 4
- Section 2 is missing similar studies that discuss metrics (like this paper)
- Metrics other than FID/R-Precision should also be discussed: MoBERT
- "Fid x, MMMD √" text should be removed from Figure 3
- Any shortcomings mentioned for proposed metrics?
- Is there an ablation study to support this claim: "Our experiments confirm that MMMD is more robust to variations in sample size" ?
- In Table 1, OTMS scores StableMoFusion (0.478) better than GT (0.481), but human judgement favors GT in Table 3 (18.93 vs. 19.68). This should be explained.
- From Eq.9, OTMS seems to be application of Sinkhorn distance (limited novelty)
- OTMS, MMMD use the new MoCLIP encoder, while FID/R-Precision use the old standard encoders. Since the paper introduces new embedding MoCLIP, it is not clear if the performance gain comes from the OT/MMD formulation or because MoCLIP is a better feature extractor.
- "𝜆= 0.02 can achieve the largest ot based Top-k A.4 value on the test set data". Does this mean hyperparameter tuning was done on test data set?

---

### Official Review · Reviewer_MZJH · 2025-12-31
**Towards Better Evaluation Metrics for Text-to-Motion Generation**

**Rating:** 6
**Confidence:** 4

**Review:**

The authors created the Optimal Transport Matching Score (OTMS) to check the global semantic alignment. The paper introduces MoCLIP to make better semantic representations for motion and text. The MMMD metric is great because it does not use a Gaussian assumption.
A few Points to Improve
• The current datasets are too small to train really smart foundation models.
• There is embedding bias that makes the AI look better than the real data.
• The old FID metric is a biased estimator and can give wrong results.
• The OTMS score changes a lot depending on the regularization parameter called lambda.
• The metrics depend too much on how good the motion encoder is.
• Scientists need to find expressive motion representations that can control specific attributes.
• The old R-Precision metric is not good because it only looks at local matching in small groups

---

### Official Review · Reviewer_i8ZD · 2026-01-07
**Strong and well-justified metrics that significantly advance evaluation for text-to-motion generation**

**Rating:** 9
**Confidence:** 4

**Review:**

This paper addresses a fundamental and increasingly critical problem in text-to-motion (T2M) generation: the misalignment between commonly reported quantitative metrics and actual motion quality as perceived by humans. The authors provide a thorough analysis of the limitations of widely used metrics such as FID and R-Precision, identifying both theoretical and empirical failure modes. To overcome these issues, the paper introduces two new evaluation metrics: Optimal Transport Matching Score (OTMS) and MoCLIP-based Maximum Mean Discrepancy (MMMD).

OTMS reformulates text–motion matching as a global optimal transport problem, moving beyond local retrieval-based evaluation, while MMMD provides a distribution-level comparison that avoids the Gaussian assumptions and estimator bias inherent in FID. Extensive experiments across multiple datasets, models, and human evaluation protocols demonstrate that the proposed metrics correlate significantly better with human judgment than existing standards.

Overall, this work makes a strong case that progress in text-to-motion generation is currently bottlenecked by flawed evaluation practices, and it offers principled, well-validated alternatives.

Strengths

High-impact problem selection: Evaluation is a core infrastructure issue for the T2M field. By directly targeting metric failures that lead to misleading conclusions (including models outperforming ground truth under existing metrics), this paper addresses a problem whose importance cannot be overstated.

Strong theoretical grounding: The critiques of FID (Gaussian assumption, bias under finite samples) and R-Precision (local ranking, embedding bias) are technically sound and supported by both theory and empirical evidence.

Well-motivated metric design:

OTMS introduces a genuinely global notion of semantic alignment via optimal transport, which is a natural and elegant formulation for batch-level text–motion matching.

MMMD provides an unbiased, distribution-free alternative to FID that is particularly well-suited to the non-Gaussian, multimodal nature of motion data.

Comprehensive empirical validation: The authors evaluate a wide range of state-of-the-art T2M models on multiple datasets, report confidence intervals over repeated runs, and include carefully designed human evaluation studies.

Strong alignment with human judgment: Correlation analyses clearly show that OTMS and MMMD outperform traditional metrics in matching human preferences, which is ultimately the gold standard for generative evaluation.

Field-shaping potential: This work has the potential to directly influence how future T2M papers report results and compare models, similar to how FID reshaped image generation evaluation.

Weaknesses

The paper is relatively dense, and some sections (particularly the discussion) could benefit from minor streamlining for readability.

While the proposed metrics are well-justified, adoption by the community will depend on the availability and standardization of MoCLIP; a brief discussion of long-term standardization could strengthen the presentation.

These issues are minor and do not detract from the overall contribution.

Overall Assessment

This is a strong and important paper that tackles a foundational issue in text-to-motion generation with both technical rigor and practical insight. The proposed metrics are theoretically principled, empirically validated, and demonstrably superior to existing standards in aligning with human perception.

In my view, this work represents exactly the kind of contribution that should be encouraged and highlighted: it does not merely incrementally improve models, but instead raises the quality bar for how progress in the field is measured. I strongly recommend acceptance.

---

### Meta-Review · Area_Chair_sBbp · 2026-01-15

**Recommendation:** Accept (Oral)
**Confidence:** 4

**Metareview:**

This paper addresses a critical bottleneck in text-to-motion generation by proposing two novel evaluation metrics, including Optimal Transport Matching Score (OTMS) and MoCLIP-based Maximum Mean Discrepancy (MMMD). The reviewers recognize the importance of this contribution, noting the paper's strong theoretical grounding, comprehensive empirical validation, and superior alignment with human judgment. The authors have provided thorough responses addressing concerns about readability, standardization of MoCLIP, hyperparameter sensitivity, and the need for expanded human evaluation. While minor concerns remain regarding the density of presentation and community adoption pathways, the authors have committed to concrete revisions including streamlined organization, public release of MoCLIP weights and evaluation toolkit, expanded statistical analysis of the human evaluation dataset, and documentation for reproducibility. Therefore, the recommendation is Accept.

---

### Decision · Program_Chairs · 2026-01-16

Accept (Oral)